# Investigation of Zearalenone Adsorption and Biotransformation by Microorganisms Cultured under Cellular Stress Conditions

**DOI:** 10.3390/toxins11080463

**Published:** 2019-08-07

**Authors:** Agnieszka Rogowska, Paweł Pomastowski, Justyna Walczak, Viorica Railean-Plugaru, Joanna Rudnicka, Bogusław Buszewski

**Affiliations:** 1Centre for Modern Interdisciplinary Technologies Nicolaus Copernicus University in Toruń, Wileńska 4, 87-100 Torun, Poland; 2Department of Environmental Chemistry and Bioanalytics, Faculty of Chemistry, Nicolaus Copernicus University in Toruń, Gagarina 7, 87-100 Torun, Poland

**Keywords:** mycotoxins, probiotic microorganisms, silver nanoparticles, MALDI-TOF MS, metabolism

## Abstract

The zearalenone binding and metabolization ability of probiotic microorganisms, such as lactic acid bacteria, *Lactobacillus paracasei*, *Lactococcus lactis,* and yeast *Saccharomyces cerevisiae,* isolated from food products, were examined. Moreover, the influence of cellular stress (induced by silver nanoparticles) and lyophilization on the effectiveness of tested microorganisms was also investigated. The concentration of zearalenone after a certain time of incubation with microorganisms was determined using high-performance liquid chromatography. The maximum sorption effectiveness for *L. paracasei*, *L. lactis,* and *S. cerevisiae* cultured in non-stress conditions was 53.3, 41.0, and 36.5%, respectively. At the same time for the same microorganisms cultured at cellular stress conditions, the maximum sorption effectiveness was improved to 55.3, 47.4, and 57.0%, respectively. Also, the effect of culture conditions on the morphology of the cells and its metabolism was examined using microscopic technique and matrix-assisted laser desorption ionization-time of flight mass spectrometry, respectively.

## 1. Introduction

Zearalenone (ZEA) is a nonsteroidal mycotoxin produced as a secondary metabolite of mold fungi from *Fusarium* family. It is commonly found in corn and small grains, such as rice, barley, wheat, or millet. ZEA contaminates feeds, as well as food, of both plant and animal origin [1,2]. The presence of zearalenone has been detected in milk, beer, oil, flour, and chocolate [2,3,4,5,6,7]. It is estimated that the average daily intake of zearalenone by adults ranges from 0.8–29 ng for 1 kg of body weight [8]. ZEA exhibits strong genotoxic and cytotoxic effect. However, the main threat underlies in its xenoestrogenic effect [9]. Therefore, as the chemical structure of ZEA is similar to natural estrogens, especially to oestradiol, it can bind to estrogen receptors and thus disturb the natural hormonal balance [10,11]. Currently conducted research has shown that the concentration of 1.0 ppm of ZEA in the pig diet may lead to hyper estrogenic syndromes, and the highest concentration may result in miscarriages and problems with conception [12]. Moreover, it has been proved that natural exposition on zearalenone can lead to changes in the female reproductive organs, such as cervical cancer [13,14]. ZEA also exhibits a strong adverse effect on animal and human after its biotransformation, which occurs mostly in the liver. The main produced metabolites are α- and β-zearalenol (α-, β-ZOL). The estrogenic potential of both metabolites is different. Form α is about 10-times more toxic than parent ZEA, while β form shows much less toxicity than ZEA [15].

Due to the adverse effect of zearalenone on human and animal health, modern science faced a huge challenge, which is the development of effective methods for its neutralization from the feed, raw materials, and food products. Nowadays, for this purpose, few physical and chemical methods are often used. However, these methods are quite expensive and may lead to loss of essential nutrients. An alternative for these methods may constitute microbiological methods, which rely on the selection of appropriate microorganisms that bind or metabolize mycotoxins without producing any harmful side effects [5]. These methods seem to be cheaper, ecofriendly, and safe in comparison to standard methods (physical and chemical methods) [16,17]. The greatest interest in the microbiological removal of mycotoxins is focused on probiotic lactic acid bacteria and some yeast species commonly used in the food industry [6]. Many of the studies carried out so far have shown that these microorganisms efficiently bind and metabolize zearalenone [6,18,19,20,21]. Moreover, differences in mycotoxin biosorption intensity by live and dead cells of the same strain has been observed. Tabari et al. [22] indicated that dead *Lactobacillus rhamnosus* and *Saccharomyces cerevisiae* cells showed greater biosorption capacity of aflatoxin B1 and ochratoxin A than the live one. However, the influence of the culture conditions of these microorganisms on the effectiveness of the neutralization process has not yet been investigated.

During life, cells of microorganisms are repeatedly subjected to various types of stress. The term “stress” can be defined as any harmful environmental factor that leads to physiological changes in cells and disturbs their homeostasis [23]. Although stressors can initially induce damage in cells and impair their functioning, prolonged exposure to such stimuli can lead to acclimation of cells through specific changes in their morphology. Cells of microorganisms are characterized by unusual adaptive ability. This is because they cannot isolate themselves from changes in the physical and chemical properties of the environment. The molecular mechanisms underlying these reactions are very diverse. However, all lead to changes in the biochemical composition of cells [23,24,25,26,27]. From the biological point of view, two types of stress can be distinguished: abiotic and biotic [23]. Abiotic stress includes all physicochemical environmental factors, such as exposure to heavy metals, xenobiotics, metal nanoparticles, radiation, changes in temperature and pH, or osmotic stress [28,29,30]. While biotic stress is associated with the interaction of cells with other organisms, such as predator-prey interactions or parasitic interactions [23,31]. Defense mechanisms that respond to environmental changes are universal. Continuous or regular exposure of a cell to a given stressor is associated with its acclimation to this factor. The cell’s response may be specific only to one stressor or to several different stressors [23,26,27]. One of the factors that can cause cellular stress and force cells to activate defense mechanisms through changes in their metabolism and morphology is their culture in the presence of metal nanoparticles and their oxides [32,33,34].

This research aimed to investigate the effect of culturing conditions of microorganisms for its ability to bind and metabolize the zearalenone. In this fact, probiotic lactic acid bacteria *Lactobacillus paracasei* and *Lactococcus lactis,* as well as yeast *Saccharomyces cerevisiae,* were used for the study. As a stress factor, silver nanoparticles synthesized by *Lactococcus lactis* strain were used. Moreover, the effect of lyophilization of cells on the effectiveness of the zearalenone adsorption process was investigated. Also, the effect of stress conditions on cells’ morphology and the metabolism was examined using microscopic and spectrometric approach.

## 2. Results

### 2.1. Influence of Culture Conditions on Zearalenone Neutralization and Metabolization Process

To investigate the ability of the studied microorganisms to neutralize and biotransform ZEA, the concentration of ZEA and its metabolites after a certain time of ZEA incubation with microbial culture was measured. Moreover, the influence of culture conditions and lyophilization on the effectiveness of these processes was determined as well. Table 1 summarizes the concentrations of ZEA determined in the supernatant solution after 30 min, 5 h, and 24 h of incubation of the microorganisms with ZEA, and Figure 1 shows the effectiveness of sorption process. It can be observed that for all the applied microbial variants, the zearalenone concentration decreased with time. These results indicated that the tested microorganisms demonstrated the ability to bind the mycotoxin, causing its neutralization. Besides, the control performed for Mueller Hinton (MH) medium treated with ZEA incubated for 30 min, 5 h, and 24 h showed that the ZEA concentration did not change along the time. This result proved that components of culture medium, such as casein hydrolysates, were not fundamental for the adsorption process. Moreover, the HPLC data showed that the conditions for growing microorganisms had a significant impact on the effectiveness of the adsorption process. For all tested microorganisms, the efficiency of neutralization after 24-h incubation was higher for cells cultured in the presence of silver nanoparticles. This phenomenon was probably related to the generation of cellular stress by silver nanoparticles and an adaptation mechanism of microbial cells. The presence of such stress factor stimulated microbial cells to activate a range of defense mechanisms, which resulted in changes in cells metabolism and morphology. As a consequence, all of these changes could affect the zearalenone adsorption and biotransformation efficiency. The difference was most pronounced in the case of yeast (*S. cerevisiae*), where the efficiency for cells grown under favorable (untreated cells with AgNPs (silver nanoparticles)) and unfavorable (treated cells) conditions was 36.5 and 57.0%, respectively (Figure 1). Moreover, *S. cerevisiae* cells cultured in the presence of LCLB56 nanoparticles were characterized by the highest neutralization efficiency among all tested microorganisms. The lyophilization also significantly influenced the ZEA adsorption effectiveness by the tested microorganisms. In the case of cells grown under favorable conditions, lyophilization caused a significant decrease in the effectiveness of the process. After 24 h of incubation, adsorption effectiveness decreased from 41.0 to 15.2, from 53.3 to 24.1, and from 36.5 to 18.5% in the case of *L. lactis*, *L. paracasei,* and *S. cerevisiae*, respectively. A similar phenomenon was observed for *L. lactis* cultured in the presence of silver nanoparticles. In turn, after lyophilization of *L. paracasei* and *S. cerevisiae* cells, the toxin neutralization efficiency significantly decreased in the initial stage of the process. However, after 24-h of incubation, the effectiveness of process intensively increased, and the final efficiency was comparable to that obtained for non-lyophilized counterparts.

Moreover, using HPLC technique, it was possible to assess the ability of the studied microorganisms to biotransform the zearalenone. Table 2 summarizes the concentrations of individual metabolites determined in tested samples. It could be observed that the tested microorganisms conducted stereoselective reduction of ZEA to α- and β-ZOL, and the culture conditions had a key role in the nature and intensity of this phenomenon (Figure 2). In the case of *L. lactis*, after 24 h incubation, biotransformation to α form was observed. However, the same bacteria cultured under cellular stress conditions were able to metabolize ZEA to both forms with the predominance of β-ZOL. After incubation of native *L. paracasei* cells with ZEA, no metabolites were observed even at long incubation times. However, it can be seen that as a result of the culture of this bacteria in the medium with addition of LCLB56 silver nanoparticles, ZEA was converted to both α- and β-ZOL. In turn, native yeast cells were able to metabolize zearalenone only to β-ZOL, while counterparts, cultured under cellular stress conditions, conducted more intensive biotransformation to β form, as well as a small amount of α-ZOL was produced. In the case of lyophilized cells, the microorganisms cultured in non-stress conditions were not able to biotransform ZEA. On the other hand, in the case of lyophilized cells cultured in the presence of silver nanoparticles, the amount of the produced metabolites was much smaller in comparison to native cells, and biotransformation was recorded only after 24 h of incubation. Exceptions were *L. paracasei* cells, where lyophilization led to an increase in the intensity of ZEA biotransformation to α-ZOL compared to the non-lyophilized counterparts.

### 2.2. Fourier Transform Infrared Spectroscopy Analysis of Microorganisms after ZEA Neutralization

To identify the surface functional groups of microorganisms that can participate in the binding of zearalenone, spectroscopic studies were carried out. These studies also enabled the tracking of changes in cells resulting from the different culture conditions of microorganisms as well as their lyophilization. The FT-IR analysis for bacteria was carried out in the range of υ = 1350–1850 cm^−1^ of amide groups region characteristic for proteins.

Figure 3D shows the FT-IR spectrum of zearalenone. On the recorded spectrum, the occurrence of a band at about υ = 1400 cm^−1^ (1), characteristic of the stretching vibrations of the hydroxyl group, could be observed. An intense band at υ = 1486 cm^−1^ (2), derived from the C-H stretching vibrations of the methyl group, could also be observed. Signals appearing in the range of 1510–1530 cm^−1^ (3,4) could be attributed to the C=C stretching vibrations in the aromatic ring of the zearalenone molecule, whereas the signals at 1648 (5), 1687 (6), and 1787 cm^−1^ (7) indicated the presence of stretching vibrations of C=O carbonyl group. These results are consistent with the chemical structure of zearalenone, which has in its structure two -OH groups attached to the aromatic ring and two carbonyl groups on the 14-membered ring of the macrocyclic lactone [21,35,36].

Figure 3; Figure 4 summarize the FT-IR spectra for all tested variants of microbial cells before and after 24-h incubation with ZEA. In case of *L. lactis* (Figure 4), it could be observed that culturing of microorganisms in the presence of silver nanoparticles resulted in more pronounced separation of signals in the range of υ = 1620–1700 and υ = 1725–1765 cm^−1^, probably originating from the C=O stretching vibrations of the carbonyl group of amides. On the other hand, the lyophilization process resulted in shallowing and overlapping of these signals compared to the spectra of non-lyophilized cells. However, incubation with zearalenone did not lead to significant changes in the FT-IR spectrum. In turn, on the spectra recorded for lyophilized counterparts, and cells cultured under stress conditions after incubation with ZEA, sharpening and better separation of the previously discussed signals were observed. In the case of lyophilized *L. lactis* cells cultured in the presence of LCLB56 nanoparticles, these signals began to overlap and slightly widen. In the last variant, the intensity of the band increased with the maximum at υ = 1449 cm^−1^, and a small signal appeared at υ = 1578 cm^−1^, which might come from the C=C stretching vibrations in the zearalenone aromatic ring.

At the spectra recorded for *L. paracasei* (Figure 3A–C) after culturing in cellular stress conditions, the shallowing and overlapping of signals at υ = 1630–1750 cm^−1^ were observed. A similar, although slightly less intense, effect on the obtained spectrum was cell lyophilization. Incubation of native bacteria with zearalenone led to a slight cleavage of signals present in the range of υ = 1405–1470 and υ = 1515–1570 cm^−1^. In the case of bacteria cultured in the presence of silver nanoparticles, incubation with ZEA led to a clear separation of signals in the range of υ = 1630–1695 and υ = 1730–1750 cm^−1^ and flattening of the signal at υ = 1405–1470 cm^−1^. In turn, for lyophilizates of both variants, as a result of incubation with the ZEA, only a small signal sharpening was observed in the ranges of υ = 1405–1470 and υ = 1730–1750 cm^−1^.

The least changes in FT-IR spectra registered for different cell variants were observed for *S. cerevisiae* (Figure 4). The cultivation of yeast in the medium with the addition of nanoparticles, as well as the lyophilization of both variants, resulted in sharpening and merging the previously delicately split band in the range of υ = 1415–1470 cm^−1^, as well as a small band narrowing in the range of υ = 1620–1695 cm^−1^. Twenty-four hours incubation of non-lyophilized cells with ZEA did not significantly affect the resulting FT-IR spectrum. However, in the case of both variants of lyophilized cells after incubation with ZEA, the signal widening in the range of υ = 1620–1695 cm^−1^ and appearance of a second maximum at υ = 1634 cm^−1^ might be observed.

### 2.3. Changes in Morphology and Metabolism of Cells Cultured in Different Conditions

To investigate the impact of culture conditions and lyophilization on the physiological condition of the tested microorganisms’ cells, the fluorescence microscope observations were conducted. Staining with acridine orange and ethidium bromide enables the differentiation of dead and living cells. Acridine orange binds to nucleic acids of both living and dead cells, whereas ethidium bromide can only bind to the DNA of dead cells, which have lost the integrity of the cell membrane. Using this approach during microscopic observation, live cells showed green fluorescence, and dead cells showed red. Figure 5 shows microscopic pictures of tested microbial cells cultured in favorable and unfavorable conditions and its lyophilizates. It can be observed that in the case of *L. paracasei*, the higher viability was demonstrated by native cells grown in De Man, Rogosa and Sharpe Broth (MRSB) medium without the addition of silver nanoparticles. However, culture in the presence of LCLB56, as well as cell lyophilization, did not significantly affect their viability. Only a small decrease in the intensity of green fluorescence of the cells could be observed. It indicated that due to unfavorable culture conditions and lyophilization, the cells began to gradually die. This phenomenon occurred more intensively in the case of *L. lactis*. Incubation with silver nanoparticles caused a slight increase in the red fluorescence of cells compared to bacteria incubated in the pure medium. Besides, the lyophilization process also significantly contributed to the decrease in cell viability. Both of *L. lactis* cells grown under favorable and unfavorable conditions after the lyophilization process began to die as evidenced by the extinction of green fluorescence and the appearance of a more yellow - transitional color characteristic of dying cells. In the case of *S. cerevisiae*, only a small decrease in cell viability after its culturing in the presence of silver nanoparticles could be observed. In turn, the lyophilization process, in this case, influenced the change in cell morphology compared to native cells. The vacuum hypertrophy and decrease in yeast viability were observed.

Furthermore, to investigate the changes in protein expression of tested microorganisms, which was a response to the adaptation to the cellular stress conditions, the MALDI-TOF MS approach was used. Figure 6 shows the obtained mass spectra for microorganisms cultured at different conditions, and Table 3 summarizes the most important *m*/*z* signals. It can be observed that on the recorded spectra for microorganisms cultured in the presence of silver nanoparticles, many significant changes appeared in comparison to the spectra recorded for counterparts grown under favorable conditions. In case of *L. paracasei,* many new signals appeared, e.g., 2696, 3145, 3825, 6750, or 10168 *m*/*z,* and one of the signals present on native bacteria spectra disappeared after incubation with silver nanoparticles (7347 *m*/*z*). A similar phenomenon was observed in the case of *L. lactis*. The appearance (e.g., 2044, 2098, 3840, 6857, 8348 *m*/*z*) and disappearance (2332 and 3084 *m*/*z*) of signals were also observed here. The most significant changes in the spectrum were observed in the case of yeast. Cultivation in the presence of nanoparticles resulted in the appearance (e.g., 3337, 3880, 8677, 13774 *m*/*z*) and disappearance (e.g., 2293, 5352, 7635, 11604 *m*/*z*) of many signals on the spectrum. These results indicated that the culture conditions had a significant influence on the expression of proteins of the tested microorganisms.

## 3. Discussion

Many recently conducted research studies have shown that both lactic acid bacteria and yeast can neutralize mycotoxins through its physical binding to the cell wall. Shetty et al. [16] have shown that *S. cerevisiae* can bind more than 60% of aflatoxin. Besides, studies conducted by Vega et al. [19] indicated that different strains of lactic acid bacteria could adsorb more than 40% of zearalenone. Niderkorn et al. [17] demonstrated that *Streptococcus* and *Enterococcus* strains were able to bind 49, 33, 24, and 62% of ZEA, deoxynivalenol (DON), fumonisin B1 (FB_1_), and fumonisin B2 (FB_2_), respectively. In our work, the effectiveness of zearalenone adsorption after 24-h of incubation with an initial concentration of 2 µg/mL was 50.0, 53.3, and 36.5% for *L. lactis*, *L. paracasei,* and *S. cerevisiae*, respectively. These results seem to be comparable to those obtained for other microorganisms in earlier studies. Moreover, the effect of bacteria concentration and viability, as well as the pH of the environment, on toxins binding have been previously studied [5,16,37,38]. Bejaouii et al. [37] demonstrated that heat-treated yeast showed higher absorption of ochratoxin A in comparison with live cells, and the number of cells had a significant influence on the process effectiveness. In turn, our studies indicated that lyophilized cells of bacteria and yeast growth in pure medium, which were characterized by lower viability, showed lower neutralization capacity of zearalenone. Lyophilization has an adverse effect, such as damage to some sensitive proteins, which leads to decreased cells viability or activity. During lyophilization, microbial cells are exposed to two types of cellular stress. The first one is freezing, and the second is drying [39,40]. Thus, lyophilization process slightly affected the viability of the cells that led to changes in the effectiveness of zearalenone adsorption after 24 h of incubation, decreasing it from 41.0 to 15.2, from 53.3 to 24.1, and from 36.5 to 18.5% for *L. lactis*, *L. paracasei,* and *S. cerevisiae*, respectively. A similar phenomenon was observed by Zang et al. [5]. They reported that post-culture filtrate and heat-inactivated yeast cells were not able to neutralize zearalenone, while live cells showed the high efficiency of this process. The results obtained by Wang et al. [38] indicated that most of the tested active LAB strains showed the better binding ability of patulin than heat-inactivated cells, suggesting that the microbial neutralization of zearalenone is likely enzymatic. Another research group has found that heat treatment of *Lysinibacillus* sp. significantly reduced the ZEA neutralization rate from 95.8 to 10.4%. They also confirmed that zearalenone reduction is strictly dependent on temperature, pH, and initial concentration of mycotoxin. The optimal conditions for ZEA reduction were a pH of 7.0 and 37 °C [41]. Many studies have also shown the effect of culture conditions, such as temperature, pH, incubation time, and growth medium, on antifungal properties of LAB [42]. However, the effect of cellular stress on the mycotoxins neutralization process have not been yet investigated. Addition of silver nanoparticles to the culture medium resulted in increased effectiveness of zearalenone adsorption from 41.0 to 47.4, from 53.3 to 55.3, and from 36.5 to 57.0% for *L. lactis*, *L. paracasei,* and *S. cerevisiae*, respectively. Although the observed differences at this stage of the study did not seem significant, the culture in the presence of nanoparticles influenced the significant increase in toxin binding efficiency after 24 h of incubation by lyophilized cells. After lyophilization of microbial cells cultured in the presence of biosilver nanoparticles, more ZEA (11.6, 30.3, and 33.9% of *L. lactis*, *L. paracasei,* and *S. cerevisiae*, respectively) compared to the cells grown in the pure medium was neutralized. Therefore, it can be assumed that the lyophilization process led to cell damage, which is consistent with the results obtained by fluorescence microscopy, and this influenced the effectiveness of ZEA adsorption. However, cells grown under unfavorable conditions could create adaptation mechanisms [43].

The Fourier transform infrared spectroscopy study showed that in the case of all tested microorganisms, the dominant share in the zearalenone binding on the surface of cells had carbonyl groups derived from the amides. Moreover, in the ZEA sorption process, π–π hydrophobic interactions between microbial cells took place, like it was described previously by Król et al. [21]. However, the surface binding process was not the only mechanism involved in the mycotoxin neutralization process in this case. The metabolism of zearalenone to α- and β-ZOL was also a significant contribution. In the literature, there are several reports on the stereoselective reduction of zearalenone as a result of bacteria and yeast metabolism. Niderkorn et al. [17] found that 11 from 202 tested LAB strains were able to metabolize ZEA, and the produced metabolite was α-ZOL. A similar study conducted by El-Sharkaway et al. [44] showed that *Streptomyces griseus* and *Streptomyces rutgersensis* were also able to reduce ZEA to α form. Besides, the ability to transform zearalenone was also described for many yeast strains. Böswald et al. [45] study indicated that *Torulaspora delbrückii, Candida tropicalis,* and *Zygosaccharomyces rouxii* could metabolize ZEA to both α- and β-ZOL. In turn, some yeast strains *Candida*, *Hansenula*, *Schizosaccharomyces*, *Brettanomyces,* and *Saccharomycopsis* genera were able to transform ZEA only to more toxic α-ZOL. In contrast to above-mentioned results, our study indicated that only native *L. paracasei* cells did not show the ability to reduce zearalenone, while *L. lactis* produced only small amounts of more toxic α-ZOL and *S. cerevisiae* transformed ZEA only to β-ZOL. Cultivation of all microorganisms in the presence of silver nanoparticles resulted in a much more intensive metabolism of ZEA to both α and β forms. However, the predominant produced metabolite was less toxic β-ZOL [15]. Lyophilization of cells in the case of those cultured in the pure medium resulted in a complete prediction of the ZEA metabolism. On the other hand, in the case of those cultured under the conditions of cellular stress, the quantities of produced metabolites after lyophilization were decidedly lower, and the dominant metabolite was still β-ZOL. The amount of produced α-ZOL was on the level of 66.09, 189.98, and 126.93 ng/mL after 24 h of incubation for *L. lactis*, *L. paracasei,* and *S. cerevisiae*, respectively. Such result would indicate that the use of freeze-dried microorganisms cultured under cellular stress conditions for the decontamination of feed and cereals would result in increased ZEA neutralization efficiency, through its binding and transformation mainly to less toxic β form with only slight production of the α-ZOL.

Cultivation of cells in the presence of a stress factor could affect permanent changes in their metabolism, which in turn could have contributed to differences in the nature and effectiveness of ZEA adsorption and biotransformation. Since microorganisms are characterized by unusual adaptive ability, they can change their metabolism depending on environmental conditions [23,24,25,26,27]. Previous studies have shown that the presence of metal nanoparticles (e.g., silver, iron) in the microbial growth environment causes morphological changes in the cell wall, as well as the formation of oxidative stress. Besides, metal nanoparticles tend to adsorb onto the surface of the bacterial cell wall, which may interfere with their ability to absorb nutrients from the environment [32,33,34]. Sacca et al. [33] showed that as a result of the exposure of *Pseudomonas stutzeri* to iron nanoparticles, both proteins involved in defense mechanisms against oxidative stress and membrane proteins were modulated. Also, FT-IR and FT-Raman study conducted by Dalai et al. [46] indicated that titanium oxide nanoparticles could change the morphology of the cells through an increase of protein and polysaccharides as a result of cell response to cell-nanoparticle interactions. MALDI-TOF MS studies showed that also in the case of tested strains, incubation with silver nanoparticles led to changes in cell morphology and metabolism. Numerous changes observed in the spectra of untreated and nanoparticle treated cells suggest that incubation with AgNPs could lead to the changes at the molecular level of membrane, cytosol, and ribosomal proteins. According to the UniProt database, the appearance of a signal at 3145 *m/z* on the spectrum of *L. paracasei* incubated with silver nanoparticles could be derived from acetyltransferase belonging to the transferase group. This protein is an enzyme enabling the transfer of functional groups from one molecule to another [47]. Moreover, a new signal at 6750 *m*/*z* might be associated with the presence of GTPase - protein regulating various aspects of processes of intracellular transport [48]. In the case of *L. lactis,* a new signal at 6857 *m*/*z* could correspond to phage protein belonging to the family of putative transcription repressor proteins [49]. There could be also observed disappearance of signaling at 2332 *m*/*z* after incubation with AgNPs, which might indicate the inhibition of major capsid protein expression. The most changes could be observed on the spectrum of *S. cerevisiae* after treatment with silver nanoparticles in comparison to those cultured in pure medium. Such numerous changes could have contributed to a significant increase in the effectiveness of adsorption, the largest of all the examined microorganisms. The new signal appeared at 3337 *m*/*z* could be related with a 60S ribosomal protein expression. This protein is a component of the ribosome, and it is responsible for the synthesis of proteins in the cell. All of these changes identified in the tested cells after its incubation with silver nanoparticles contributed to more efficient binding or transfer to the interior of the cell, as well as the effect on the metabolism of zearalenone in comparison to neutralization capacity of cells grown in non-stress culture conditions.

## 4. Materials and Methods

### 4.1. Chemicals and Reagents

Zearalenone, α-zearalenol, β-zearalenol, and other chemicals, i.e., α-cyano-4-hydroxycinnamic acid, formic acid, ethanol, dimethyl sulfoxide, acetonitrile, ethidium bromide, phosphate-buffered saline, and acridine orange were provided by Sigma-Aldrich (St. Louis, MO, USA).

### 4.2. Microorganisms

The *Lactobacillus paracasei* and *Lactococcus lactis* strains were isolated from dairy products (Dairy Cooperative in Drzycim, Poland), according to Milanowski et al. [50]. The 16S rDNA nucleotide sequences of *L. paracasei* LB3 and *L. lactis* 56 have been submitted in GenBank and deposited in the Polish Collection of Microorganisms (PCM) under deposit no. B/00146 and B/00116, respectively. The *Saccharomyces cerevisiae* was isolated from baker’s yeast (Gliwice, Poland) and identified by 28 S rDNA, internal transcribed spacer (ITS) sequencing, and using MALDI-TOF MS technique with BioTyper software, according to previous methodology [51]. *L. paracasei*, *L. lactis,* and *S. cerevisiae* were cultured in De Man-Rogosa-Sharpe (MRS) Broth, Tryptic Soy (TS) Broth, and Yeast Extract-Peptone-Dextrose (YPD) Broth sterile medium, respectively (Sigma Aldrich, Buchs, Switzerland).

### 4.3. Culturing of Microorganisms under Cellular Stress Conditions

As a stressor agent, the silver nanoparticles LCLB56 obtained by microbiological synthesis using *Lactococcus lactis* source was used. Their physicochemical characteristics were described previously by Railean-Plugaru et al. [52].

The tested microorganisms were cultured in sterile culture medium with addition of LCLB 56 silver nanoparticles at concentrations of 10% of the minimal inhibitory concentration value (6.25, 12.50, and 50.00 µg/mL) with the final concentration 0.625, 1.250, and 5.000 µg/mL for *L. paracasei*, *L. lactis,* and *S. cerevisiae*, respectively. After 72 h, the culture was transferred to sterile falcon tubes and centrifuged (15 °C, 7000 rpm, 5 min). Obtained bacteria pellet was two times washed by sterile water and stored at −80 °C; part of them was lyophilized (lyophilizer FeeZone, Labconco, Kansas City, MO, USA).

### 4.4. Zearalenone Neutralization and Metabolization

Bacterial pellets and lyophilizates cultured, according to Section 4.3, were suspended in 10 mL of sterile Mueller Hinton minimal liquid medium. The optical density of each sample was 6.00 McFarland (18 × 10^8^ CFU/mL). The prepared solution of zearalenone in DMSO was added to each tube resulting in a final concentration of 2 μg/mL, then, incubated at 37 °C for 24 h.

### 4.5. High-Performance Liquid Chromatography Analysis

All of the samples obtained, as described in Section 4.4, were subjected to the extraction of zearalenone (ZEA, α-ZOL, and β-ZOL). Two milliliters of the obtained sample incubated at a certain interval (30 min, 5 h, and 24 h) was transferred to a sterile Eppendorf tube and centrifuged (RT, 5 min, 14,000 rpm). The collected supernatant (1 mL) was extracted twice with 500 μL of chloroform. The obtained chloroform phases were combined and evaporated to dryness [53]. One milliliter ethanol was well mixed with each evaporated sample and placed to an ultrasonic bath (30 min, 40 °C). The prepared extract was analyzed by high-performance liquid chromatography (HPLC).

The concentration of ZEA, α-ZOL, and β-ZOL was measured using Shimadzu HPLC-ESI-MS/MS 8050 (Tokyo, Japan). The instrument includes a controller (CBM 20A), a binary solvent delivery system (LC-30AD), an autosampler (SIL-30A), and a column thermostat (CTO-20AC). Instrument control, data acquisition, and processing were performed with LabSolution 5.8 software for HPLC. MS/MS detection was operated in negative ion mode in the mass range of *m*/*z* 150–1000. The electrospray ionization (ESI) settings were as follows: nebulizing gas flow 3 L/min, heating gas flow 10 L/min, temperature of the drying gas 400 °C, and interface temperature 300 °C. ZEA and metabolites were separated using a Synergi Hydro-RP column dimensions of 150 × 2.0 mm with a pore size of 80 Å (Phenomenex, Torrance, CA, USA), flow rate of 0.2 mL/min, injection volume of 1 µL, separation temperature of 25 °C, and the isocratic mobile phase consisting of 50% of acetonitrile and 50% of 0.1% formic acid in water. ZEA and metabolites were monitored in the scheduled multiple reaction monitoring (MRM) mode. MRM transitions were for ZEA: 317.2→131.2, α-ZOL, and β-ZOL: 319.25→130.1, collision energy was 35 eV.

All of the tested samples were prepared and analyzed in triplicate. For the obtained results, the arithmetic averages and standard deviations were calculated.

### 4.6. FT-IR Spectroscopic Analysis

An aliquot (2 mL) of untreated and treated samples (bacterial pellet and lyophilizates) with ZEA (described in 2.4.) after 24 h of incubation was centrifuged (RT, 5 min, 14,000 rpm). The obtained pellets were washed with sterile distilled water. Two microliters of samples were dropped on the Assay-free card and allowed to dry [54,55]. The untreated cells and methanol solution of ZEA served as a blank. The FT-IR spectra were recorded in the range of v = 1350–1850 cm^−1^ at room temperature using Direct Detect spectrometer (Merck, Darmstadt, Germany).

### 4.7. Microscopic Analysis

Bacterial pellets were suspended in sterile water. Then, ethidium bromide (λ_exc_ = 493 nm, λ_em_ = 620 nm, final concentration 0.4 μg/mL) and acridine orange (λ_exc_ = 503 nm, λ_em_ = 530/540 nm, final concentration 0.12 μg/mL) were added. After 5 min of incubation, the sample was centrifuged (4000 rpm, 4 min), and the obtained bacterial pellet was resuspended in phosphate-buffered saline (PBSx1). The viability and morphology of the cells were analyzed using fluorescence microscope Zeiss Axiocom D1 (Göttingen, Germany) with Axio Vision 4.8. software and set of filters 43 He and 38.

### 4.8. Matrix-Assisted Laser Desorption/Ionization with Mass Spectrometry Analysis

Bacterial pellets were washed with physiological salt and centrifuged (13,000 rpm, 4 min). Next, pellets were suspended in 900 μL of ethanol and 300 μL of water and centrifuged (13,000 rpm, 2 min). The obtained bacterial pellets were suspended in 2 μL of acetonitrile and 2 μL of 70% formic acid. Then, samples were centrifuged (13,000 rpm, 2 min), and 1 μL of each sample was spotted on MALDI-TOF-MS Ground Steel Plate in three repetitions. After drying, 1 μL of α-cyano-4-hydroxycinnamic acid (HCCA) matrix at a concentration of 10 mg/mL was spotted on each spot and also dried. MS spectra were recorded in reflective positive ionization mode in the 2000–20000 *m*/*z* range on Ultraflex Extreme II spectrometer with smart beam laser (λ = 355 nm, 2 kHz frequency). For the evaluation of spectrometric data, FlexControl and FlexAnalysis software were used.

## 5. Conclusions

The present study showed that the lactic acid bacteria, *L. paracasei*, *L. lactis,* and the yeast *S. cerevisiae* could neutralize zearalenone. The selection of appropriate culture conditions for microorganisms, such as the addition of substances that cause cellular stress to the culture medium, affects the efficiency of neutralization and the ability to metabolize zearalenone. Cells of all tested microorganisms cultured in the medium with the addition of silver nanoparticles showed better neutralization capacity of ZEA and improved its metabolism to α- and β-ZOL with a definite predominance of form, which is practically harmless to human and animal health. The lyophilization process also has a significant impact on the effectiveness of the adsorption of the test mycotoxin and causes its significant reduction in the case of microorganisms cultured in the traditional way. On the other hand, lyophilization of cells cultured in the presence of cellular stress did not affect the ZEA binding intensity after 24 h of the process. Moreover, the microscopic study also confirmed that the culturing method and the lyophilization process had a significant impact on the physiological state and morphology of the studied microorganisms. Besides, MALDI-TOF MS study showed that growth of tested microorganisms in the presence of silver nanoparticles in the medium had a significant influence on their molecular profiles.

## Figures and Tables

**Figure 1 toxins-11-00463-f001:**
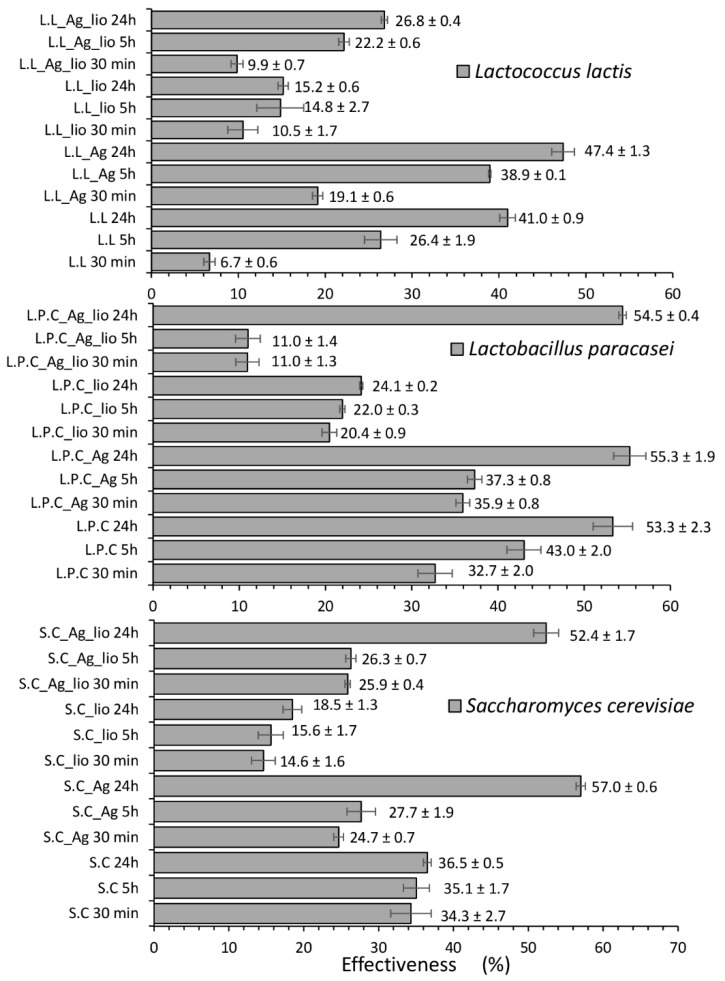
Effectiveness of zearalenone adsorption by *L. lactis* (L.L), *L. paracasei* (L.P.C), and *S. cerevisiae* (S.C) cultured in untreated growth medium and in medium treated with silver nanoparticles (L.L_Ag, L.P.C_Ag, S.C._Ag), as well as their lyophilizates (“lio”) after 30 min, 5 h, and 24 h of incubation.

**Figure 2 toxins-11-00463-f002:**
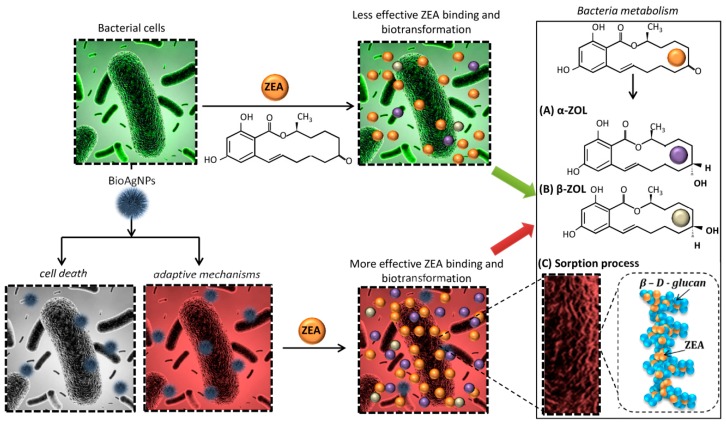
Proposed mechanism of zearalenone (ZEA) neutralization process by microbial cells grown in untreated growth medium and treated medium with biologically synthesized silver nanoparticles.

**Figure 3 toxins-11-00463-f003:**
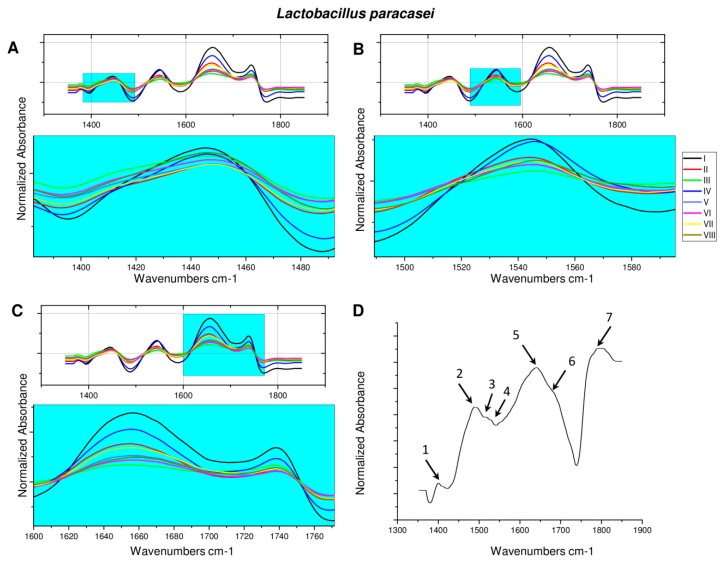
FT-IR spectrum of *L. paracasei* (**A**–**C**): I—bacteria cells (control); II—cells treated with zearalenone (ZEA); III—cells treated with silver nanoparticles (AgNPs); IV—cells treated with AgNPs_ZEA; V—lyophilized cells; VI—lyophilized cells treated with ZEA; VII—lyophilized cells treated with AgNPs; VIII—lyophilized cells treated with AgNPs_ZEA; zearalenoene (**D**): 1—υ = 1400 cm^−1^; 2—υ = 1486 cm^−1^; 3,4—υ = 1510-1530 cm^−1^; 5—υ = 1648 cm^−1^; 6—υ = 1687 cm^−1^; 7—υ = 1787 cm^−1^.

**Figure 4 toxins-11-00463-f004:**
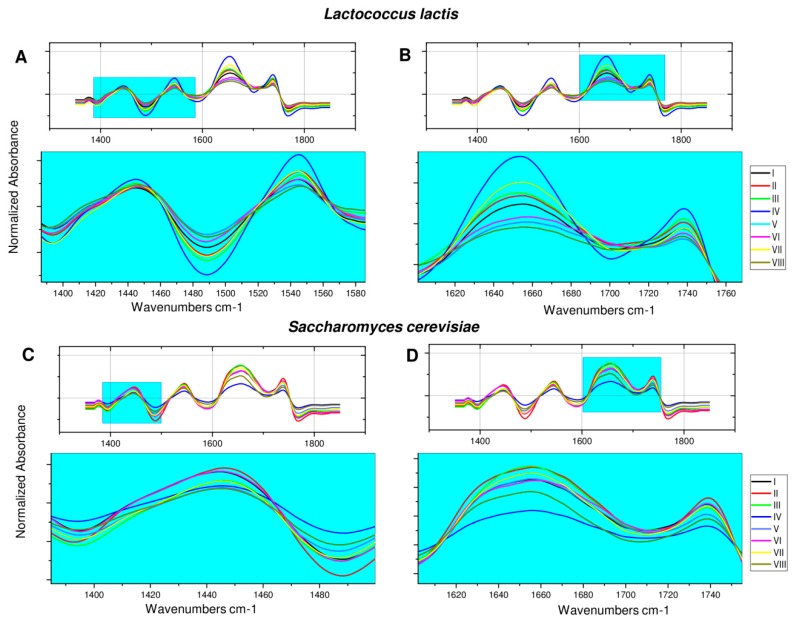
FT-IR spectrum of *L. lactis* (**A**,**B**) and *S. cerevisiae* (**C**,**D**): I—microbial cells (control); II—cells treated with zearalenone (ZEA); III—cells treated with AgNPs; IV—cells treated with AgNPs_ZEA; V—lyophilized cells; VI—lyophilized cells treated with ZEA; VII—lyophilized cells treated with AgNPs; VIII—lyophilized cells treated with AgNPs_ZEA.

**Figure 5 toxins-11-00463-f005:**
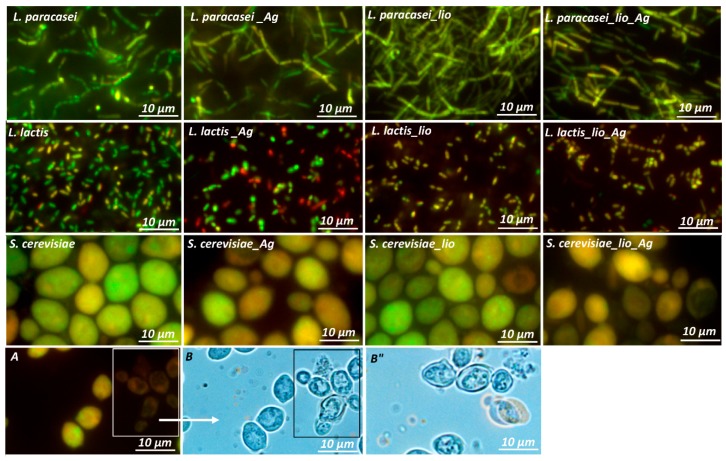
Microscopic image of *L. lactis*, *L. paracasei,* and *S. cerevisiae* cells cultured in untreated growth medium and treated medium with silver nanoparticles (with note “Ag”), as well as their lyophilizates (with note “lio”): A—cells viability; B, B”—vacuolation of cells.

**Figure 6 toxins-11-00463-f006:**
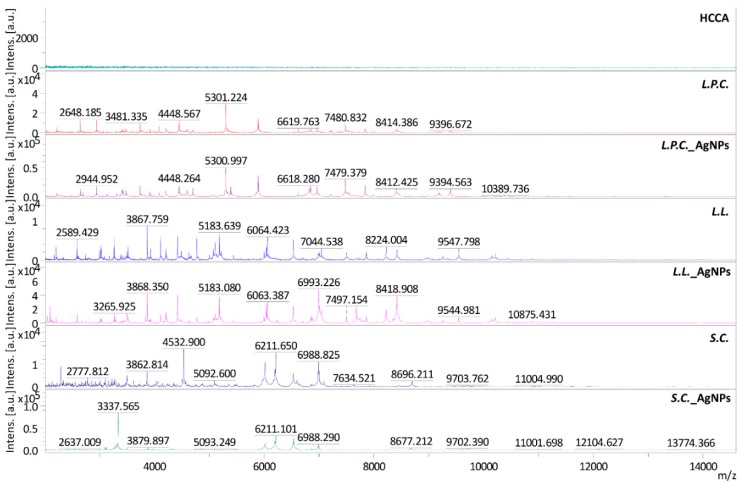
MALDI-TOF MS spectra of α-cyano-4-hydroxycinnamic acid (HCCA) matrix, *L. lactis* (L.L.), *L. paracasei* (L.P.C.)*,* and *S. cerevisiae* (S.C.) cultured in untreated growth medium and treated medium with silver nanoparticles (AgNPs).

**Table 1 toxins-11-00463-t001:** Comparison of zearalenone (ZEA) concentrations in the extract obtained from the supernatant after incubation of *L. lactis* (L.L), *L. paracasei* (L.P.C), and *S. cerevisiae* (S.C.) cells grown under non-stress and stress conditions (L.L_Ag, L.P.C_Ag, S.C._Ag – treated with silver nanoparticles), as well as their lyophilizates (“lio”) with zearalenone for 30 min, 5 h, and 24 h.

	C_ZEA_ ± SD [μg/mL]			
**t = 0 min**	2.09 ± 0.02			
	***Lactococcus lactis***			
**Variant**	**L.L.**	**L.L._Ag**	**L.L._lio**	**L.L._lio_Ag**
t = 30 min	1.95 ± 0.01	1.69 ± 0.01	1.87 ± 0.04	1.88 ± 0.01
t = 5 h	1.54 ± 0.04	1.27 ± 0.00	1.78 ± 0.06	1.63 ± 0.01
t = 24 h	1.23 ± 0.02	1.10 ± 0.03	1.77 ± 0.01	1.53 ± 0.01
	***Lactobacillus paracasei***			
**Variant**	**L.P.C.**	**L.P.C._Ag**	**L.P.C._lio**	**L.P.C._lio_Ag**
t = 30 min	1.40 ± 0.04	1.34 ± 0.02	1.66 ± 0.02	1.86 ± 0.03
t = 5 h	1.19 ± 0.04	1.31 ± 0.02	1.63 ± 0.01	1.86 ± 0.03
t = 24 h	0.97 ± 0.05	0.93 ± 0.04	1.59 ± 0.00	0.95 ± 0.01
	***Saccharomyces cerevisiae***			
**Variant**	**S.C.**	**S.C._Ag**	**S.C._lio**	**S.C._lio_Ag**
t = 30 min	1.37 ± 0.06	1.57 ± 0.01	1.78 ± 0.03	1.55 ± 0.01
t = 5 h	1.35 ± 0.04	1.51 ± 0.04	1.76 ± 0.03	1.54 ± 0.01
t = 24 h	1.32 ± 0.01	0.90 ± 0.01	1.70 ± 0.03	1.00 ± 0.04

**Table 2 toxins-11-00463-t002:** Comparison of α- and β-ZOL (zearalenol) concentrations in the supernatant after incubation of *L. lactis* (L.L), *L. paracasei* (L.P.C), and *S. cerevisiae* (S.C) cells grown under favorable and unfavorable (L.L_Ag, L.P.C_Ag, S.C._Ag – treated with silver nanoparticles) conditions, as well as their lyophilizates (“lio”) with zearalenone for 30 min, 5 h, and 24 h.

	C _α-ZOL_ ± SD [µg/mL]				C _β-ZOL_ ± SD [µg/mL]			
	***Lactococcus lactis***							
**Variant**	**L.L.**	**L.L._Ag**	**L.L._lio**	**L.L._lio_Ag**	**L.L.**	**L.L._Ag**	**L.L._lio**	**L.L._lio_Ag**
t = 30 min	-	-	-	-	-	-	-	-
t = 5 h	-	0.107 ± 0.004	-	-	-	1.097 ± 0.041	-	-
t = 24 h	0.048 ± 0.002	0.262 ± 0.006	-	0.066 ± 0.002	-	1.464 ± 0.020	-	0.356 ± 0.019
	***Lactobacillus paracasei***							
**Variant**	**L.P.C.**	**L.P.C._Ag**	**L.P.C._lio**	**L.P.C._lio_Ag**	**L.P.C.**	**L.P.C._Ag**	**L.P.C._lio**	**L.P.C._lio_Ag**
t = 30 min	-	-	-	-	-	-	-	-
t = 5 h	-	-	-	-	-	-	-	-
t = 24 h	-	0.142 ± 0.004	-	0.190 ± 0.008	-	0.699 ± 0.054	-	0.310 ± 0.024
	***Saccharomyces cerevisiae***							
**Variant**	**S.C.**	**S.C._Ag**	**S.C._lio**	**S.C._lio_Ag**	**S.C.**	**S.C._Ag**	**S.C._lio**	**S.C._lio_Ag**
t = 30 min	-	-	-	-	-	-	-	-
t = 5 h	-	-	-	-	0.034 ± 0.000	0.012 ± 0.002	-	-
t = 24 h	-	0.152 ± 0.002	-	0.127 ± 0.004	0.102 ± 0.017	0.857 ± 0.013	-	0.423 ± 0.013

**Table 3 toxins-11-00463-t003:** Summary of signals observed on MALDI-TOF MS spectra where (+) means signals that appeared and (−) signals that disappeared after incubation with nanoparticles.

*m/z*	*L. paracasei*	*m/z*	*L. lactis*	*m/z*	*S. cerevisiae*
2696	+	2044	+	2293	−
3145	+	2098	+	3337	+
3825	+	2332	−	3863	−
5393	+	3084	−	3880	+
6750	+	3840	+	4324	+
7347	−	6234	+	4534	−
7647	+	6857	+	5352	−
8997	+	6993	+	7091	−
10168	+	7678	+	7635	−
10390	+	8348	+	8677	+
				8696	−
				11604	−
				12105	+
				13774	+
				14639	+

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
