# Peer review of "Investigation of Zearalenone Adsorption and Biotransformation by Microorganisms Cultured under Cellular Stress Conditions"

_toxins, 2019, doi:10.3390/toxins11080463_

Round 1

Reviewer 1 Report

The manuscript is an interesting study that investigated the effect of silver nanoparticles on the adsorption and biotransformation of zearalenone by some microorganisms. It is concluded that the cultivation of those microorganisms in the presence of silver nanoparticles improve their capacity to adsorb and biotransform ZEA. This effect was more pronounced in the yeast S. cerevisiae. The study was conducted with care and rigor. The manuscript is well structured, but some sentences or paragraphs need a revision of English. Concerning to results, table 1 and figure 1 need a statistical treatment to verify if differences are significantly different from controls. Bellow I also recommend other modification/corrections. In the attached document authors can found all the comments and corrections needed. For these reason I recommend the publication of this manuscript with major revision.

Title: Investigation of Zearalenone Neutralization by Microorganisms Cultured under Cellular Stress Conditions.

The term “Neutralization” is not the better form to define the effects on ZEA. I recommend to change to “adsorption and biotransformation”.

Line 5: “The ability to bind and metabolize zearalenone by …”

Change to:  “The zearalenone binding and metabolization ability of …”

Line 7: “… cellular stress …”

Change to: “… cellular stress (induced by silver nanoparticles) …”

Line 8: “… culture effectiveness of the tested microorganisms the …”

Change to: “… on the effectiveness of tested microorganisms the …”

Line 11: “… good conditions …”

Change to: “… non-stress conditions …”

Change in all the manuscript similar situations

Line 13: “… 55.27, 47.36 and 57.01%.”

Change to: “… was improved to 55.3, 47.4 and 57.0%, respectively.”

Use only one decimal case in all the manuscript

Line 38: “… parent ZEA while β form is practically harmless [7,15,16].”

please check if this idea is totally correct. As far as I know beta form has approx. half the toxicity of ZEA. Please see:

Shier WT, Shier AC, Xie W, Mirocha CJ. 2001. Structure-activity relationships for human estrogenic activity in zearalenone mycotoxins. Toxicon. 9//;39:1435-1438.

[7,15,16] are not the best reference to support this statement.

Line 83-84: “… microorganisms to neutralization and biotransformation of zearalenone,”

Change to: “… microorganisms to neutralize and biotransform zearalenone,”

Line 97-98: “ … was higher for cells cultured in the presence of silver nanoparticles.”

You can explain why this happened?

Line 100: “ … was 36.51% and 57.01%, respectively.”

They are statistically different?

Figure 2:

They all look the same. If possible, enlarge the image.

Line 210: “ … caused a significant increase …”

How you can say it is significant you didn´t do a statistical analysis.

Figure 3:

You need to label the image A and B as the one between both.

Line 260-261:

Revise English of the sentence. Its meaning is not clear.

Author Response

Response to Reviewer 1 Comments

manuscript no.: toxins-548916

"Investigation of zearalenone adsorption and biotransformation by microorganisms cultured under cellular stress conditions"

Dear Editor and Reviewers,

We would like to thank the editor and reviewers for their criticism and helpful remarks that will help us to improve our work. Please find below the replies, point by point, to each comments.

In manuscript file all of the changes have been provided in red color.

Reviewer 1:

Point 1: The manuscript is an interesting study that investigated the effect of silver nanoparticles on the adsorption and biotransformation of zearalenone by some microorganisms. It is concluded that the cultivation of those microorganisms in the presence of silver nanoparticles improve their capacity to adsorb and biotransform ZEA. This effect was more pronounced in the yeast S. cerevisiae. The study was conducted with care and rigor. The manuscript is well structured, but some sentences or paragraphs need a revision of English. Concerning to results, table 1 and figure 1 need a statistical treatment to verify if differences are significantly different from controls. Bellow I also recommend other modification/corrections. In the attached document authors can found all the comments and corrections needed. For these reason I recommend the publication of this manuscript with major revision.

Response 1: According to Reviewer suggestion the error bards and standard deviation for each samples were added in the Figure 1.

Point 2: Title: Investigation of Zearalenone Neutralization by Microorganisms Cultured under Cellular Stress Conditions.

The term “Neutralization” is not the better form to define the effects on ZEA. I recommend to change to “adsorption and biotransformation”.

Response 2: Taking into consideration the Reviewer suggestion the title was changed from "Investigation of zearalenone neutralization by microorganisms cultured under cellular stress conditions" to "Investigation of zearalenone adsorption and biotransformation by microorganisms cultured under cellular stress conditions".

Point 3: Line 5: “The ability to bind and metabolize zearalenone by …”

Change to:  “The zearalenone binding and metabolization ability of …”

Line 7: “… cellular stress …”

Change to: “… cellular stress (induced by silver nanoparticles) …”

Line 8: “… culture effectiveness of the tested microorganisms the …”

Change to: “… on the effectiveness of tested microorganisms the …”

Line 11: “… good conditions …”

Change to: “… non-stress conditions …”

Change in all the manuscript similar situations

Line 13: “… 55.27, 47.36 and 57.01%.”

Change to: “… was improved to 55.3, 47.4 and 57.0%, respectively.”

Use only one decimal case in all the manuscript

Response 3: All of the above changes have been introduced in the manuscript as was suggested by the Reviewer.

Point 4: Line 38: “… parent ZEA while β form is practically harmless [7,15,16].”

please check if this idea is totally correct. As far as I know beta form has approx. half the toxicity of ZEA. Please see:

Shier WT, Shier AC, Xie W, Mirocha CJ. 2001. Structure-activity relationships for human estrogenic activity in zearalenone mycotoxins. Toxicon. 9//;39:1435-1438.

[7,15,16] are not the best reference to support this statement.

Response 4: Taking into account the Reviewer suggestions and remarks, the authors have used a more appropriate reference in the respective sentence. Moreover, the information about β-ZOL toxicity has been changed according to results obtained by Shier et al. (2001) as was suggested by the Reviewer. It was resigned from the statement "non-toxic" and changed it to "less toxic".

Point 5: Line 83-84: “… microorganisms to neutralization and biotransformation of zearalenone,”

Change to: “… microorganisms to neutralize and biotransform zearalenone,”

Response 5: According to the Reviewer suggestion, the manuscript has been improved by changing the ”….microorganisms to neutralization and biotransformation of zearalenone,” to “…microorganisms to neutralize and biotransform zearalenone,”

Point 6: Line 97-98: “ … was higher for cells cultured in the presence of silver nanoparticles.”

You can explain why this happened?

Response 6: This phenomenon is probably related with the generation of cellular stress by silver nanoparticles and adaptation mechanism of microbial cells. The presence of such stress factor stimulates microbial cells to activate a range of defense mechanisms, which results in changes in cells metabolism and morphology. As a consequence, all of these changes can affect on the zearalenone adsorption and biotransformation efficiency.

Considering the Reviewer remark the manuscript has been supplied with  additional explanation regarding the increase in the zearalenone binding efficiency of cells in the presence of silver nanoparticles.

Point 7: Line 100: “ … was 36.51% and 57.01%, respectively.”

They are statistically different?

Response 7: The authors have written like this, because obtained standard deviation are relatively low (36.51 ± 0.51 and 57.01 ± 0.62%) and has been decided that they are statistically different. In order to avoid any doubt and to make the results clearer to the reader, Figure 1 was supplemented with standard deviation values for each sample, as was suggested by Reviewer.

Point 8: Figure 2: They all look the same. If possible, enlarge the image.

Response 8: According to the Reviewer remark, the image has been improved and manuscript supplemented with the new images. 

Point 9: Line 210: “ … caused a significant increase …”

How you can say it is significant you didn´t do a statistical analysis.

Response 9: Yes, is true, the author did not provide the description of statistical analysis. The authors have based by the increasing of red fluorescence in case of the cells treated with silver nanoparticles. We are agree with the Reviewer remark and therefore, the manuscript has been improved by using "....caused a slight increase in the red fluorescence of cells".

Point 10: Figure 3: You need to label the image A and B as the one between both.

Response 10: The differences between them is the using of different filters for imaging in order to represent the vacuolation of S. cerevisiae under stress conditions. Taking into consideration the Reviewer remark the Fig. 3 (present Figure 5) has been supplemented with the missing information.

Point 11: Line 260-261: Revise English of the sentence. Its meaning is not clear.

Response 11: According to the Reviewer remark, the sentence: “Thus, in the resin partial damage or cell death after the lyophilization process effectiveness of zearalenone neutralisation after 24 h of incubation decrease from…” has been changed to the “Thus, lyophilization process slightly affected the cells viability what led to changes in effectiveness of zearalenone adsorption after 24 h of incubation that decrease from…”

Reviewer 2 Report

The manuscript “Investigation of Zearalenone Neutralization by Microorganisms Cultured under Cellular Stress Conditions”, submitted to Toxins, is a well done research and well written manuscript. However, I think that there are several points that the authors should better clarify to improve the understanding of the paper. Furthermore, this study is quite poor regarding to use of new and efficient citations. Please, see below my specific comments.

TITLE

OK

ABSTRACT

OK

KEYWORDS

Line 16: Please, use words that are not already present in the title.

INTRODUCTION

Line 23: I prefere “is a nonsteroidal”

Line 28: Use the acronym ZEA for “Zearalenone”

Line 35: Use the acronym ZEA for “Zearalenone”

Line 51: you appoint for the first time L. rhamnosus and S. cerevisiae in the main text, please use the full name

Lines 63 - 65: I think that you should add some references, as examples, to support this sentence “Abiotic stress includes all physicochemical environmental factors such as exposure to heavy metals, xenobiotics, metal nanoparticles, radiation, changes in temperature and pH, or osmotic stress.” I would like to suggest:

Li, H., et al. (2015). The impact of temperature on microbial diversity and AOA activity in the Tengchong Geothermal Field, China. Scientific reports, 5, 17056.

Hechmi, N., et al. (2016). Depletion of pentachlorophenol in soil microcosms with Byssochlamys nivea and Scopulariopsis brumptii as detoxification agents. Chemosphere, 165, 547-554.

Zhang, L., et al. (2017). Characteristics of rumen microorganisms involved in anaerobic degradation of cellulose at various pH values. RSC Advances, 7(64), 40303-40310.

RESULTS

Line 84: Line 35: Use the acronym ZEA for “Zearalenone”

Line 95: Delete the reference from this point. In this section you must only show you results not compare/discuss it with other studies

Lines 113 – 115: All the acronyms should be explained in the table caption (eg. L.L. L.L._Ag L.L._lio L.L._lio_Ag).

Lines 116 – 119: Please, add the label under the x-axis

Lines 140 – 141: All the acronyms should be explained in the table caption (eg. L.L. L.L._Ag L.L._lio L.L._lio_Ag).

Line 149: Line 95: Delete the reference from this point. In this section you must only show you results not compare/discuss it with other studies

Lines 161 – 164: Lines 116 – 119: Please, add the label close to the y-axis

Line 169: Delete the reference from this point. In this section you must only show you results not compare/discuss it with other studies

Line 177: Delete the reference from this point. In this section you must only show you results not compare/discuss it with other studies

Line 203: Delete the reference from this point. In this section you must only show you results not compare/discuss it with other studies

Lines 220 - 221: What parts of the microorganisms am I seeing?

DISCUSSION

Lines 311 – 314: You should move this figure in the “RESULTS” section

MATERIALS AND METHODS

Line 354: Replaced “microorganisms” to “Biological Material”

Lines 404 – 407: I think that you should add some references, as examples, to support your method “An aliquot (2mL) of untreated and treated samples (bacterial pellet and lyophilizates) with ZEA (described in 2.4.) after 24h of incubation were centrifuged (RT, 5 min, 14000 rpm). The obtained pellets have been washed with sterile distilled water. 2μL of samples were dropped on the Assay-free card and allowed to dry.” I would like to suggest:

Bosso, L., et al., (2015). Biosorption of pentachlorophenol by Anthracophyllum discolor in the form of live fungal pellets. New biotechnology, 32(1), 21-25

Daneshvar, E., et al., (2017). A comparative study of methylene blue biosorption using different modified brown, red and green macroalgae–Effect of pretreatment. Chemical Engineering Journal, 307, 435-446

Lines 428 – 429: how did you analyze your data from a statistical point of view? Please add a paragraph in the “Materials and methods” section

CONCLUSION

Line 431: For the microorganisms you should use the abbreviation name

Author Response

Response to Reviewer 2 Comments

manuscript no.: toxins-548916

"Investigation of zearalenone adsorption and biotransformation by microorganisms cultured under cellular stress conditions"

Dear Editor and Reviewers,

We would like to thank the editor and reviewers for their criticism and helpful remarks that will help us to improve our work. Please find below the replies, point by point, to each comments.

In manuscript file all of the changes have been provided in red color.

Reviewer 2:

The manuscript “Investigation of Zearalenone Neutralization by Microorganisms Cultured under Cellular Stress Conditions”, submitted to Toxins, is a well done research and well written manuscript. However, I think that there are several points that the authors should better clarify to improve the understanding of the paper. Furthermore, this study is quite poor regarding to use of new and efficient citations. Please, see below my specific comments.

TITLE

OK

ABSTRACT

OK

KEYWORDS

Point 1: Line 16: Please, use words that are not already present in the title.

Response 1: According to Reviewer suggestion the keywords was changed from: “zearalenone; probiotic microorganisms; cellular stress; neutralization; metabolism” to: mycotoxins; probiotic microorganisms; silver nanoparticles; MALDI-TOF MS; metabolism”.

INTRODUCTION

Point 2: Line 23: I prefere “is a nonsteroidal”

Line 28: Use the acronym ZEA for “Zearalenone”

Line 35: Use the acronym ZEA for “Zearalenone”

Line 51: you appoint for the first time L. rhamnosus and S. cerevisiae in the main text, please use the full name

Response 2: The manuscript has been improved with the changes suggested by the Reviewer.

Point 3: Lines 63 - 65: I think that you should add some references, as examples, to support this sentence “Abiotic stress includes all physicochemical environmental factors such as exposure to heavy metals, xenobiotics, metal nanoparticles, radiation, changes in temperature and pH, or osmotic stress.” I would like to suggest:

Li, H., et al. (2015). The impact of temperature on microbial diversity and AOA activity in the Tengchong Geothermal Field, China. Scientific reports, 5, 17056.

Hechmi, N., et al. (2016). Depletion of pentachlorophenol in soil microcosms with Byssochlamys nivea and Scopulariopsis brumptii as detoxification agents. Chemosphere, 165, 547-554.

Zhang, L., et al. (2017). Characteristics of rumen microorganisms involved in anaerobic degradation of cellulose at various pH values. RSC Advances, 7(64), 40303-40310.

Response 3: Taking into consideration the Reviewer suggestion, authors has added the proposed references to support above mentioned sentence.

RESULTS

Point 4: Line 84: Line 35: Use the acronym ZEA for “Zearalenone”

Response 4: The word “Zearalenone” was changed to “ZEA” as Reviewer suggested.

Point 5: Line 95: Delete the reference from this point. In this section you must only show you results not compare/discuss it with other studies

Response 5: According to the Reviewer suggestion the reference from this point was deleted.

Point 6: Lines 113 – 115: All the acronyms should be explained in the table caption (eg. L.L. L.L._Ag L.L._lio L.L._lio_Ag).

Response 6: Taking into consideration the Reviewer suggestion all of the used acronyms was explained in the table caption.

Point 7: Lines 116 – 119: Please, add the label under the x-axis

Response 7: The label under the x-axis was added, as Reviewer suggested.

Point 8: Lines 140 – 141: All the acronyms should be explained in the table caption (eg. L.L. L.L._Ag L.L._lio L.L._lio_Ag).

Response 8: According to the Reviewer suggestion, all of the used acronyms was explained in the table caption.

Point 9: Line 149: Line 95: Delete the reference from this point. In this section you must only show you results not compare/discuss it with other studies

Response 9: The reference from this point was deleted in accordance to Reviwer suggestion.

Point 10: Lines 161 – 164: Lines 116 – 119: Please, add the label close to the y-axis

Response 10: The label of y-axis was added, as Reviewer suggested.

Point 11: Line 169: Delete the reference from this point. In this section you must only show you results not compare/discuss it with other studies

Line 177: Delete the reference from this point. In this section you must only show you results not compare/discuss it with other studies

Line 203: Delete the reference from this point. In this section you must only show you results not compare/discuss it with other studies

Response 11: According to the Reviwer suggestion, the mentioned references were deleted.

Point 12: Lines 220 - 221: What parts of the microorganisms am I seeing?

Response 12: Considering the Reviewer remark, the legend of the Figure 3 (present Figure 5) has been improved by specifying what exactly was investigated on this part of the assay.

Corrected version of the legend:

Figure 5. Microscopic image of L. lactis, L. paracasei and S. cerevisiae cells cultured in untreated growth medium and in treated medium with silver nanoparticles (with note “Ag”) as well as their lyophilizates (with note “lio”): A – cells viability; B, B” – vacuolation of cells.

DISCUSSION

Point 13: Lines 311 – 314: You should move this figure in the “RESULTS” section

Response 13: The figure 5 (present Figure 2) was moved in to “Results” section as was suggested by the Reviewer.

MATERIALS AND METHODS

Point 14: Line 354: Replaced “microorganisms” to “Biological Material”

Response 14: The header title has been changed as it was suggested by the Reviewer.

Point 15: Lines 404 – 407: I think that you should add some references, as examples, to support your method “An aliquot (2mL) of untreated and treated samples (bacterial pellet and lyophilizates) with ZEA (described in 2.4.) after 24h of incubation were centrifuged (RT, 5 min, 14000 rpm). The obtained pellets have been washed with sterile distilled water. 2μL of samples were dropped on the Assay-free card and allowed to dry.” I would like to suggest:

Bosso, L., et al., (2015). Biosorption of pentachlorophenol by Anthracophyllum discolor in the form of live fungal pellets. New biotechnology, 32(1), 21-25

Daneshvar, E., et al., (2017). A comparative study of methylene blue biosorption using different modified brown, red and green macroalgae–Effect of pretreatment. Chemical Engineering Journal, 307, 435-446

Response 15: Taking into consideration the Reviewer suggestion, the authors have added the proposed references to support above mentioned method.

Point 16: Lines 428 – 429: how did you analyze your data from a statistical point of view? Please add a paragraph in the “Materials and methods” section

Response16:  All of the tested samples were prepared and analyzed in triplicate. For the obtained results the arithmetic averages and standard deviations was calculated. In response to the Reviewer suggestions has been added this information in separate paragraph of section 4.5.

CONCLUSION

Point 17: Line 431: For the microorganisms you should use the abbreviation name

Response 17: According to Reviewer suggestion, in the manuscript has been used the abbreviation name for the microorganisms.

Round 2

Reviewer 1 Report

The authors have made the requested changes and have considerably improved the understanding of the manuscript.

Author Response

Response to Reviewer 1 Comments

manuscript no.: toxins-548916

"Investigation of zearalenone adsorption and biotransformation by microorganisms cultured under cellular stress conditions"

Reviewer 1:

Point 1: The authors have made the requested changes and have considerably improved the understanding of the manuscript.

Response 1: We would like to thank Reviewer for his comments. The manuscript was edited by native English speaker, as it was suggested by Reviewer.
